# Plasma Electrolytic Polishing—An Ecological Way for Increased Corrosion Resistance in Austenitic Stainless Steels

**DOI:** 10.3390/ma15124223

**Published:** 2022-06-14

**Authors:** Viera Zatkalíková, Štefan Podhorský, Milan Štrbák, Tatiana Liptáková, Lenka Markovičová, Lenka Kuchariková

**Affiliations:** 1Department of Materials Engineering, Faculty of Mechanical Engineering, University of Žilina, Univerzitná 8215/1, 010 26 Žilina, Slovakia; milan.strbak@fstroj.uniza.sk (M.Š.); tatiana.liptakova@fstroj.uniza.sk (T.L.); lenka.markovicova@fstroj.uniza.sk (L.M.); lenka.kucharikova@fstroj.uniza.sk (L.K.); 2Institute of Production Technologies, Faculty of Materials Science and Technology, Slovak University of Technology in Bratislava, Jána Bottu č. 2781/25, 917 24 Trnava, Slovakia; stefan.podhorsky@stuba.sk; 3Research Centre UNIZA, University of Žilina, Univerzitná 8215/1, 010 26 Žilina, Slovakia

**Keywords:** austenitic stainless steel, plasma electrolytic polishing, corrosion resistance, electrochemical impedance spectroscopy, potentiodynamic polarization

## Abstract

Plasma electrolytic polishing (PEP) is an environment-friendly alternative to the conventional electrochemical polishing (EP), giving optimal surface properties and improved corrosion resistance with minimum energy and time consumption, which leads to both economic and environmental benefits. This paper is focused on the corrosion behavior of PEP treated AISI 316L stainless steel widely used as a biomaterial. Corrosion resistance of plasma electrolytic polished surfaces without/with chemical pretreatment (acid cleaning) is evaluated and compared with original non-treated (as received) surfaces by three independent test methods: electrochemical impedance spectroscopy (EIS), potentiodynamic polarization (PP), and exposure immersion test. All corrosion tests are carried out in the 0.9 wt.% NaCl solution at a temperature of 37 ± 0.5 °C to simulate the internal environment of a human body. The quality of tested surfaces is also characterized by optical microscopy and by the surface roughness parameters. The results obtained indicated high corrosion resistance of PEP treated surfaces also without chemical pretreatment, which increases the ecological benefits of PEP technology.

## 1. Introduction

Polished stainless-steel surfaces are strictly required for a number of technical and medical applications due to their higher corrosion resistance and increased resistance to microorganisms. There are various polishing processes commonly used to obtain a smooth material surface. Traditional mechanical polishing may result in a deformed layer and residual stresses on the treated surface [1,2,3]. Conventional EP ensures very low surface roughness, without residual surface tensions and with improved corrosion resistance, but it requires strong, concentrated inorganic acids and their mixtures.

Although the plasma electrolytic polishing (PEP) procedure has been known for some decades and it has many advantages compared to electropolishing, insufficient attention has been paid to this process until now [4,5]. PEP is based on electrolysis; but, unlike EP, it proceeds at a high voltage in low concentrated, neutral non-toxic salt solutions [5,6,7,8]. Under the influence of a direct current at the voltage 220–450 V, a thin film (vapor–plasma envelope) is generated above the treated surface (polished specimen is connected to the positive pole of the power source as an anode). The electrolyte acts as a medium that conducts the electric current to the vapor–plasma envelope, and it creates appropriate conditions for its stability. The electrons are moving toward the anode at a high speed and their collisions with neutral particles produce new electrons and ions. The number of electrons increases and a discharge channel with high-density charged particles is formed. The collisions between the treated surface and electrons moving at a high speed in the discharge channel causes a removal of the material and a gradual smoothing of the surface [5,6,9,10].

In the PEP process, chemical reactions between the vapor–plasma envelope environment and the metal surface are undesirable. Therefore, electrolyte consumption does not occur and the process parameters are stable for a long time [5]. In the case of electrochemical polishing, the process reducing the surface roughness is diametrically different. The electrolyte consisting of highly concentrated acids reacts with the polished metal surface and the selective dissolution takes place [6,11,12,13].

Compared to the traditional EP, PEP technology brings high machining efficiency and, due to the high uniform polishing quality, applicability for the stainless steel work-pieces with complex and irregular shapes [5,6,9,10,14]. The above-mentioned attributes of PEP technology indicate its appropriate use in the surface treatment of austenitic stainless-steel biomaterials for implants and medical instruments that require high quality polished surfaces excellently resistant to corrosion and to bacterial attachment [14,15,16,17,18,19].

According to the available literature, the quality of PEP treated stainless steel surfaces is most often evaluated on the basis of the achieved gloss and surface roughness [4,6,7]. Studies evaluating the corrosion properties of plasma electrolytic polished surfaces and comparing them with electropolished stainless steels surfaces are lacking.

AISI 316L is a Cr-Ni-Mo stainless steel widely recommended as a biomaterial in various medical applications including long-term endoprostheses [15,16,17].

Despite the high resistance to uniform corrosion, austenitic stainless steels are prone to local pitting corrosion in halide (especially chlorides) containing environments. This corrosion form, typical for passivating metals and alloys, is initiated by aggressive ions present in the solution that penetrate through the weakened places of surface passive film and cause its local breakdown [20,21].

The objective of this research is the evaluation of the effect of plasma electrolytic polishing on the corrosion resistance of AISI 316L stainless steel used as an experimental material. The corrosion properties of PEP treated surfaces, with/without chemical pre-treatment, are evaluated and compared with original non-treated (as received) surfaces by three independent test methods: electrochemical impedance spectroscopy (EIS), potentiodynamic polarization, and exposure immersion test. All corrosion tests are carried out under conditions simulating the internal environment of a human body (0.9 wt.% NaCl solution, 37 ± 0.5 °C). The quality of tested surfaces is also characterized by digital optical microscopy and by the surface roughness parameters.

## 2. Materials and Methods

The material used was an AISI 316L austenitic stainless-steel sheet of 1.5 mm thickness, with a 2B surface finish (smooth and matte metallic glossy surface). The chemical composition obtained by X-ray fluorescence is given in Table 1.

The microstructure of the experimental material (Figure 1) observed by optical microscope is created by polyhedral austenitic grains with observable twins, which could be created by annealing or by rolling.

For PEP surface treatment, the rectangular specimens 15 mm × 40 mm were prepared. A part of specimens was chemically pre-treated by pickling (acid cleaning) according to the recommendations of authors [17] for pre-treatment of electrochemically polished coronary stents. The pickling solution was composed of hydrofluoric acid, nitric acid, and demineralized water. Volumes of the used compounds and the pickling conditions are summarized in Table 2.

PEP was carried out in specialized laboratory of the Institute of Production Technologies (Slovak University of Technology Bratislava, Faculty of Materials Science and Technology in Trnava, Slovakia) in 6 wt.% ammonium sulfate electrolyte under the following conditions: voltage 260 V, temperature 68 °C, and polishing time 3 min [5]. Then, the polishing specimens were rinsed with demineralized water and freely dried.

For comparison, corrosion resistance was also tested on the as received steel surface, i.e., the original surface without additional mechanical and chemical treatments. An overview of the tested surface conditions and the specimen designations for the experiments is described in Table 3.

A sodium chloride solution (0.9 wt.%, specific conductivity 15.55 mS/cm, pH 7.15) and the temperature of 37 ± 0.5 °C to simulate the internal environment of the human body was used as the corrosion environment for corrosion tests. All chemical compounds used in experiments were analytical grade.

The electrochemical corrosion tests (EIS and potentiodynamic polarization) were performed in the conventional three-electrode cell system with a calomel reference electrode (SCE, +0.248 V vs. SHE at 20 °C) and a platinum auxiliary electrode (Pt) using a BioLogic corrosion measuring system, with a PGZ 100 measuring unit. The time for potential stabilization between the specimen and the electrolyte was set to 10 min. The exposed area of a specimen was 1 cm^2^ (15 mm × 40 mm specimen was externally attached to a 1 cm^2^ “window” on the corrosion cell).

Electrochemical impedance spectroscopy measurements were recorded at the corrosion potential over a frequency range from 100 KHz to 5 mHz, obtaining 10 points per decade and applying a 10 mV R.M.S amplitude. Results of EIS measurements were displayed as the Bode and the Nyquist plots. The representative Nyquist curve was selected from three measurements for the same type of surface conditions. The EIS parameter values were obtained by EC-LAB software analysis of Nyquist curves.

The potentiodynamic polarization curves were recorded at the sweep rate of 1 mV/s [22,23], a potential scan range was applied between −0.30 and 1.2 V vs. open circuit potential (OCP). At least three experiment repeats were carried out for each type of surface (surface condition) and the representative curve was selected.

The specimen shape for a 50-day exposure immersion test was rectangular (15 mm × 40 mm). The specimens were degreased by ethanol and weighted out with accuracy ± 0.00001 g before the test. The group of three parallel specimens was tested for each type of surface. After exposure, the specimens were carefully brushed, washed by de-mineralized water, freely dried, and weighted out again [23]. The solution for the immersion test was stationary, without circulation. Its pH value was measured before the test.

The microstructure of the experimental material was performed by a Neophot 32 optical microscope (OP). An Olympus DSX1000 digital optical microscope (DOM) was used to image the tested surface conditions in the form of a QBQ design. The roughness parameters measurements were performed by a Mitutoyo SJ 400 Roughness Tester. For each surface type, the roughness profile in the central part of the specimen (longitudinal direction) was extracted.

## 3. Results and Discussion

The topography of the tested surfaces displayed by a digital optical microscope is shown in Figure 2. Danilov et al. [8] documented a very similar appearance of a PEP treated AISI 304 stainless steel surface. According to these authors, imperfections as dark spots observed on the surface may be undissolved inclusions related to the chemical composition of the material.

Marked differences between the as received and both polished surfaces were reflected in the surface roughness evaluated by the following roughness parameters (Table 4): arithmetical mean deviation of the assessed profile (R_a_), the average maximum peak to valley height (R_z_), and the root mean square slope (R∆q). The R∆q parameter is recommended for the description of the real roughness of the electropolished surfaces because its insensitivity to the scale is unlike the commonly used roughness amplitude parameters [24]. According to the presented results, the roughness of both plasma-polished surfaces is very similar. This is consistent with the topography of PEP and PPEP surfaces (Figure 2b,c). Moreover, low roughness parameter values point to the resistance of PEP and PPEP surfaces to bacterial attachment and to biofilm initiation [18,25,26]. If compared to the surface roughness after EP, the authors [25,27] reported approximately the same R_a_ values of the EP austenitic steel surfaces. The authors [28] presented the same R∆q values for the 316L surfaces after EP was performed at 40 °C.

### 3.1. Electrochemical Impedance Spectroscopy

For the impedance spectra measured on the tested surfaces, only one time constant was observed and therefore a single loop circuit (Figure 3) consisting of electrolyte resistance (R_Ω_), charge transfer resistance (R_ct_), and the constant phase element (CPE) was used for the evaluation.

The same circuit was also used for stainless steels by authors [29,30,31,32]. If the “*n*” exponent appearing in the mathematical relation expressing the CPE is equal to one, the CPE represents the capacitor. The CPE value depends on parameters related to the rate of ongoing processes on the electrode surface, e.g., surface roughness, different thickness, or coating’s composition [33]. According to the authors [34], the capacitance dispersion on solid electrodes is due to surface disorder (i.e., heterogeneities) and due to the roughness (i.e., geometric irregularities) much larger than those on the atomic scale. The dependence of impedance modulus |Z| and phase angle Φ on the measured frequency is shown in the form of Bode plots for different surface modifications, as shown in Figure 4.

From the overall point of view, the character of the surfaces immersed in 0.9 wt.% NaCl solution revealed similar behavior. This may be attributed to the fact that corrosion mechanisms, which take place on the passive film formed on variously treated 316L stainless steel surfaces, are of the same nature. The trend of Bode plots for all three surfaces in this study confirmed similar results obtained in the works [35,36], for the same stainless steel. The authors [36] stated that the impedance is measured toward very low frequency, without a resistive region (horizontal line associated with phase angle Φ~0°), which could be discerned over this frequency range [36,37]. In the high frequency region, 10^3^–10^5^ impedance data for all surfaces had approximately the same character, without significant changes. Whereas, from the mid to the low frequency region 10^3^–10^−2^, the impedance for PEP and PPEP surface treatment differed from the as received surface by a higher impedance value. Universally, in Bode plots, a higher impedance modulus |Z| at a lower frequency region implies better corrosion resistance of the bare metal [38]. Moreover, in the low frequency region 10^−1^–10^−2^ the curve for the as received surface started to gradually reduce its inclination. In other words, the impedance curve started to be less dependent on the frequency. PEP and PPEP surfaces were almost identical in terms of impedance evolution in all measured frequency ranges. Similarly, as in the case of impedance development, curves describing the phase angle for PEP and PPEP surfaces evolved without significant changes. However, in the final stage of the low frequency region, the inclination of the PEP impedance curve was higher than for the PPEP surface. The evolution of the phase angle for the as received surface was different during the whole frequency range, except for its highest and lowest value where the curves intersected, whereas the lowest value for phase angle Φ = 0° was measured at the high frequency range 10^4^–10^5^ for all surfaces. This behavior is also confirmed in the study [37] in which authors obtained the same results for 316L stainless steel where the absolute impedance curve at high frequencies (10^4^–10^5^ Hz) was almost independent of the frequency with a phase angle of 0°, which represents electrolyte resistance. The highest phase angle value for all surfaces was obtained in the frequency region approximately 10^1^ with its value in the interval between −70° < Φ < −80°. In general, when the absolute value of the phase angle Φ rises, a higher corrosion resistance together with better corrosion protection of the substrate are expected [39].

The impedance responses represented by Bode plots are consistent with their trends shown by the Nyquist plots, as displayed in Figure 5. The values of EIS parameters are listed in Table 5.

According to the obtained R_ct_ values (the higher R_ct_ value points to the higher quality of the passive film), plasma electrolytic polishing brought more than a six-fold increase in the quality of passive surface film in comparison to the as received surface (531.4 is approximately 6-fold compared to 87.9). Chemical pre-treatment showed only a slight positive effect on the resulting quality of the polished surface (PPEP surface). Close R_ct_ values of PEP and PPEP surfaces (531.4 kΩ·cm^2^ and 546.5 kΩ·cm^2^) agree with the close roughness parameters R_a_, R_z_, R∆q (Table 4), and they reflect their high quality and corrosion resistance. The high passive surface quality of both PEP and PPEP surfaces could be related to the mechanism and the parameters of applied PEP process, which ensured the optimal heat generation and evaporation of the metal particles from the surface relief [5,6,7,14]. This process led to optimal surface polishing, and a smooth surface with very low roughness parameters resistant to the adsorption and penetration of the aggressive chloride anions was reached. The passive film quality may be also improved by chromium enrichment (increase of Cr/Fe ratio in the passive film) [11,17].

The authors [29] evaluated the corrosion resistance of EP treated biomaterial (316L) in Hank’s solution, and they obtained a significantly lower R_ct_ value (134 kΩ·cm^2^). This means that an expensive and environment-unfriendly EP process brought a markedly lower surface quality in comparison to the PEP process. The authors [28] studied the corrosion properties of 316L stainless steel surfaces electropolished in the same solution (orthophosphoric acid + sulphuric acid + water) but at the different conditions (temperature, current density, time). The R_ct_ value closest to PEP and PPEP surfaces was 559.3 kΩ·cm^2^ obtained for specimen EP treated at 40 °C for 10 min at 0.8 A/cm^2^. Taking into account that the active components of the electrolyte are consumed by the EP process (it means the electrolyte must be changed after each specimen), it is clear that a comparable surface quality was obtained significantly more advantageously by a PEP process.

### 3.2. Potentiodynamic Polarization

The PP curves of tested surfaces are shown in Figure 6 and the values of the electrochemical PP parameters are listed in Table 6. The shape of polarization curves is typical for passivating metals; the anodic branches of all surfaces are typically passive and it points to the control of the anodic dissolution rate by the passive current density [40]. For each curve, the E_corr_ value was determined as the potential of the transition from the cathodic to the anodic branches. In the case of passivating metals, it is important to compare the width of the passive regions and the fluctuations of the passive current density and to determine the pitting potential E_p_ as the potential of a sudden permanent increase in current density to assess and to compare corrosion resistance. The pitting potential denotes the disruption of the passive surface film in the passivity region and the onset of stable pit growth. The higher E_p_ value means a higher resistance to pitting [41].

According to the E_p_ values, both PEP and PPEP treated surfaces proved to have resistance to pitting that was more than twice as high as the as received surface. For electropolished 316L stainless steel in the physiological solution, the PP curves with an extremely wide passivity region were also presented by the authors [23]. However, in this case the high corrosion resistance shown by potentiodynamic curves required a mixture of strong inorganic acids either for electropolishing or for chemical pretreatment. Contrary to that, Ghanavati et al. [29] recorded depassivation at E_p_ = 0.376 V on the potentiodynamic polarization curve of EP austenitic stainless steels and E_corr_ = −0.51 V.

Although the PEP and the PPEP curves (Figure 6) are similar, there are some differences between them. A lower E_corr_ value of the PEP surface (−0.06 V vs. SCE) points to a lower thermodynamic stability. On the other hand, the E_p_ value of the PEP surface (1.08 V vs. SCE) is higher than the E_p_ value of the PPEP surface (0.98 V vs. SCE). The passive current densities of both PEP and PPEP surfaces can be assessed and compared in Figure 7, showing the detail of polarization curves in linear axes. If taken into account that the value 0.05 mA/cm^2^ is usually considered the boundary current density between the passivity and the activity states [23,41], it is clear that PEP and PPEP processes ensured a stable passive state with a passive current density not higher than 0.005 mA/cm^2^. Fluctuations of the passive current density on both curves that could denote the metastable pit growth in the passivity region [41,42] are very slight. Differently, there is a sharp peak indicating the metastable pit growth in the passivity region of the curve for the as received surface condition (Figure 6). Based on these facts, the high resistance to the pitting of both plasma-polished surfaces can be supposed.

The use of (NH_4_)_2_ SO_4_ solution as an electrolyte for the plasma polishing process may have contributed to the high corrosion resistance of both PEP and PPEP surfaces. According to the authors [43,44], the passivation effect of sulfate ions plays an important role in the formation of a highly resistant passive film. The authors [45] explain the increase in resistance of the austenitic stainless steel to the pitting by an adsorption of sulfate anions on the passive film surface.

Achievement of high corrosion resistant plasma electrolytic polished surfaces, without the chemical pre-treatment recommended before EP, could be related to the different mechanisms of both the polishing processes. The EP process is based on the electrochemical reaction between the electrolyte and the polished metal surface, which results in a selective dissolution of the metal. Pickling (acid cleaning) before EP contributes to removal of surface oxides and contaminants [17], and it makes the following polishing process more effective. The PEP process based on the collisions between the treated surface and the high-speed electrons ensures a more even material removal, and this could be the reason for the excellent result without a chemical pre-treatment.

### 3.3. Exposure Immersion Test

The surfaces after 50 days’ exposure of tested specimens under conditions simulating the internal human body environment (0.9 wt.% NaCl solution at 37 °C) are shown in Figure 8. The average corrosion rates calculated from the mass losses of the specimens (mass loss per unit area per unit time, g/(m^2^·day)) are listed in Table 7. According to the comparison of PEP and PPEP surfaces before/after exposure immersion testing (Figure 2b,c and Figure 8b,c), there are minimal differences between them. With regard to the very low average corrosion rates (Table 7), only slight pitting corrosion attack related to the structural imperfections (e.g., inclusions visible in Figure 8) could be initiated. It points to the high corrosion resistance and biomedical safety with the minimum metal release into the human body environment [16,46,47] that correlates with the results of the performed independent electrochemical corrosion tests. Unlike these results, in the exposure test performed under the same conditions on the EP surface of 316L stainless steel, the authors [28] observed more pronounced pitting corrosion damage and higher corrosion rates (dependent on EP conditions 0.013–0.024 g/(m^2^·day)).

The received surface (Figure 8a) showed the lower quality of the passive film, which enabled the local penetration of the chloride anions in numerous places and the initiation of pitting corrosion. This is consistent with the results of both electrochemical corrosion tests.

## 4. Conclusions

PEP process performed in 6 wt.% ammonium sulfate electrolyte (voltage 260 V, temperature 68 °C, polishing time 3 min) provided a mirror finish of 316L stainless steel without the use of aggressive inorganic acids;Plasma polished surfaces showed close values of R_a_ and R∆q roughness parameters to those obtained by traditional EP;According to the obtained R_ct_ values (R_ct_ = 531.4 and 546.5 kΩ·cm^2^, respectively), plasma electrolytic polishing ensured a more than six-fold increase in the quality of the passive surface film compared to the as received surface (R_ct_ = 87.9 kΩ·cm^2^).Potentiodynamic polarization curves for both PEP and PPEP treated surfaces showed broad passivity regions and high E_p_ values (E_p_ = 1.081 and 0.98 V vs. SCE, respectively). This points to the high resistance of PEP and PPEP treated surfaces to the pitting corrosion in potentiodynamic polarization.Based on the PP measurement’s positive shift of corrosion potential E_corr_ for PEP, PPEP surface treatments against as-received surface was analyzed (+90mV for PEP and +140mV for PPEP). This indicates a positive effect of the investigated surface treatment on the thermodynamic stability of the surface.Plasma electrolytic polished surfaces remained almost unchanged during the 50-day exposure test. This agrees with the extremely low corrosion rates calculated from the mass losses (0.0003 g/(m^2^·day)).

Performed independent corrosion tests proved high corrosion resistance and slight non-essential differences between the quality of PEP and PPEP treated surfaces. According to the obtained results, plasma electrolytic polishing is not only environment-friendly alternative to the electropolishing but it does not need the chemical pre-treatment recommended in traditional EP. This means further saving of aggressive chemicals and it increases the benefits of PEP technology.

## Figures and Tables

**Figure 1 materials-15-04223-f001:**
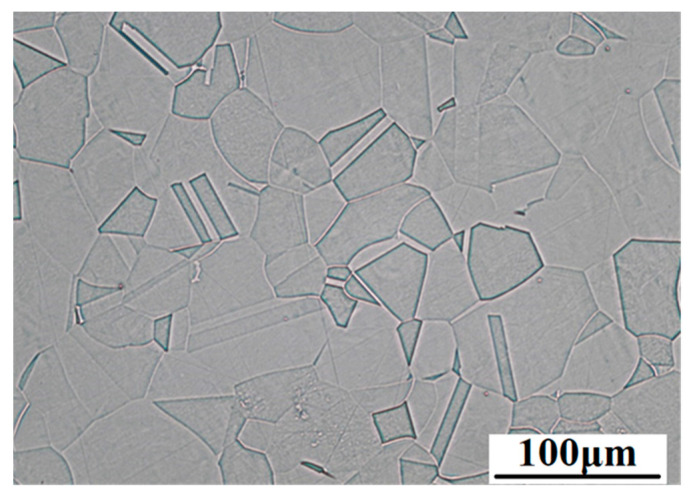
Microstructure of AISI 316L stainless steel, longitudinal section (Kalling’s 2 etch., OM, magnification 500×).

**Figure 2 materials-15-04223-f002:**
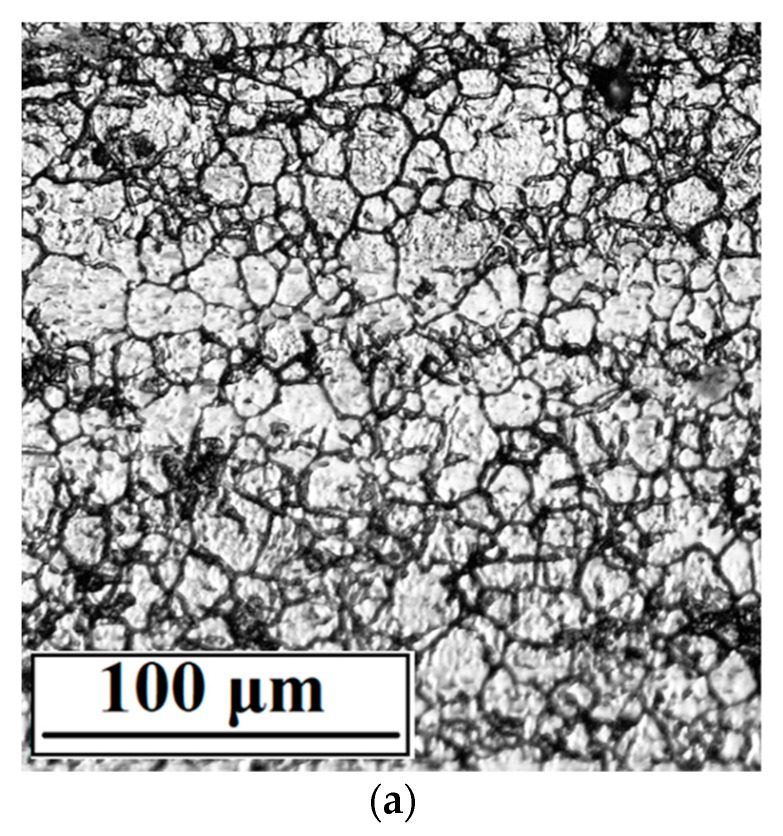
DOM surface morphology micrographs of surfaces (**a**) as received, (**b**) PEP, (**c**) PPEP (QBQ, magnification 600×).

**Figure 3 materials-15-04223-f003:**
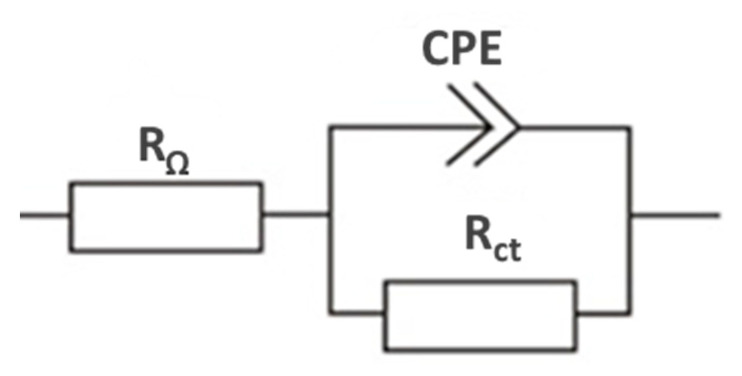
Equivalent circuit.

**Figure 4 materials-15-04223-f004:**
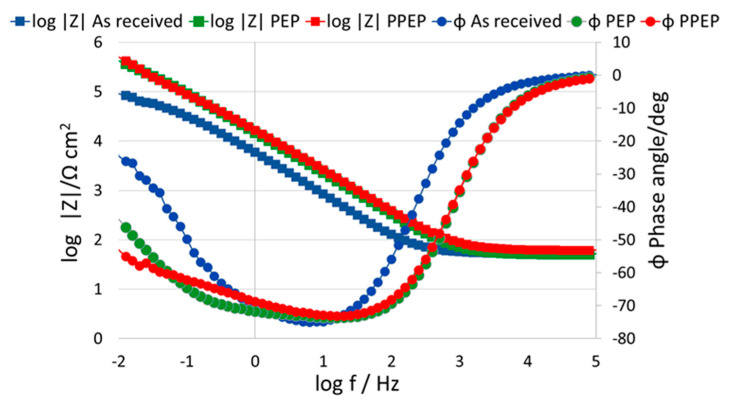
Bode plots for tested surface conditions.

**Figure 5 materials-15-04223-f005:**
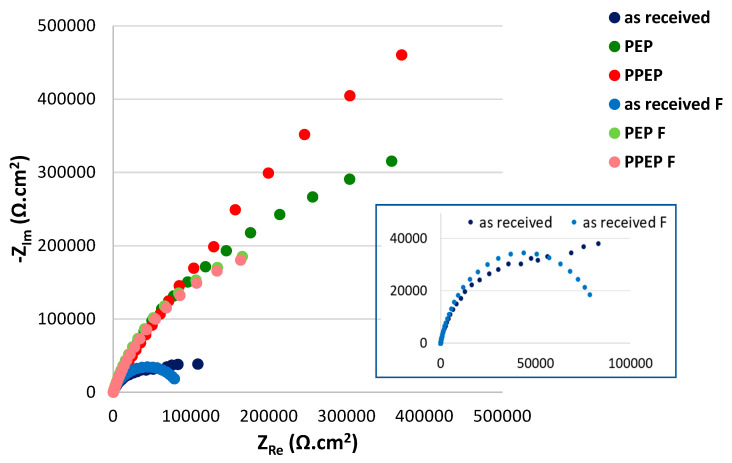
Nyquist plots for tested surface conditions (fitted curves are “F” marked).

**Figure 6 materials-15-04223-f006:**
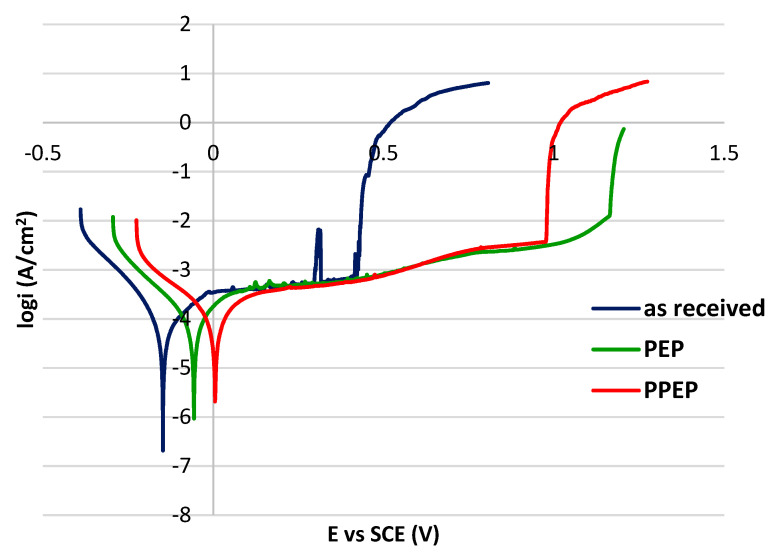
Potentiodynamic polarization curves for tested surface conditions.

**Figure 7 materials-15-04223-f007:**
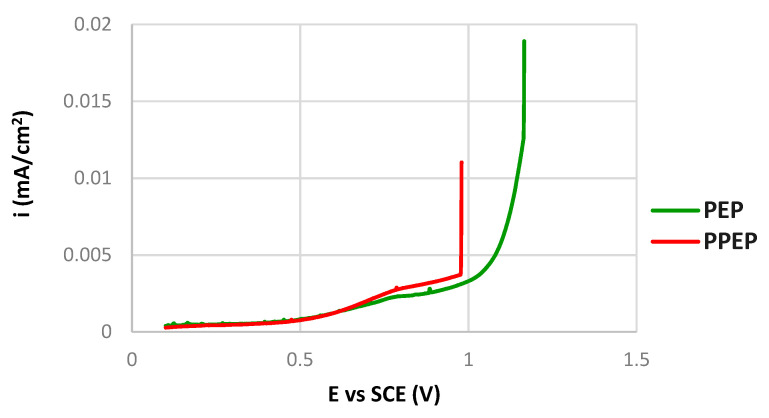
Detail of potentiodynamic polarization curves for tested surfaces in linear axes.

**Figure 8 materials-15-04223-f008:**
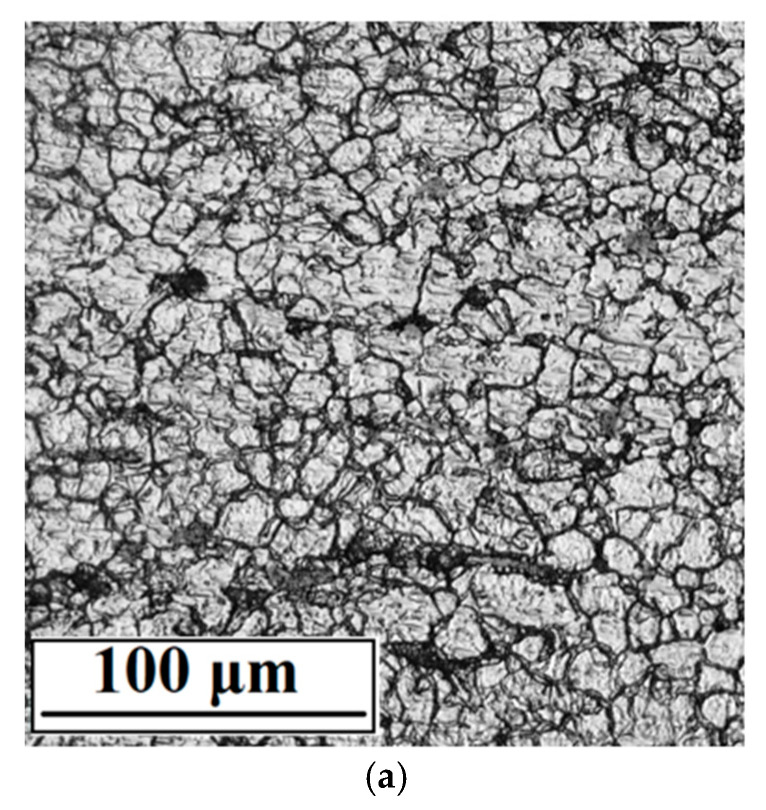
DOM surface morphology micrographs of surfaces after a 50-day exposition in 0.9 wt.% NaCl solution: (**a**) as received, (**b**) PEP, (**c**) PPEP (QBQ, magnification 600×).

**Table 1 materials-15-04223-t001:** Chemical composition of AISI 316L stainless steel (wt.%).

Cr	Ni	Mo	Mn	N	C	Si	P	S	Fe
16.79	10.14	2.03	0.82	0.05	0.02	0.031	0.03	0.001	balance

**Table 2 materials-15-04223-t002:** Conditions of pickling (acid cleaning).

Component	Volume (mL)	Temperature (°C)	Time (s)
**HF**	3	22 ± 3	3600
**HNO_3_**	9		
**H_2_O**	to 100 mL		

**Table 3 materials-15-04223-t003:** Overview of tested surface conditions.

Type of Surface	Specimen Designation
Plasma electrolytic polished	PEP
Pickled and plasma electrolytic polished	PPEP
Original non-treated	As received

**Table 4 materials-15-04223-t004:** Roughness parameter values of tested surface conditions: arithmetical mean deviation (R_a_), the average maximum peak to valley height (R_z_), the root mean square slope (R∆q).

Specimen Designation (Type of Surface)	R_a_ (µm)	R_z_ (µm)	R∆q (-)
as received	0.22	2.30	0.15
PEP	0.11	1.00	0.04
PPEP	0.10	1.00	0.04

**Table 5 materials-15-04223-t005:** Values of EIS parameters.

Specimen Designation (Type of Surface)	Charge Transfer Resistance R_ct_ (kΩ·cm^2^)	CPE (μF/cm^2^)	Exponent n	Electrolyte Resistance R_Ω_ (kΩ·cm^2^)
as received	87.90 ± 0.4	37.00 ± 0.19	0.85 ± 0.002	0.05 ± 0.002
PEP	531.40 ± 1.2	14.51 ± 0.11	0.83 ± 0.003	0.05 ± 0.002
PPEP	546.50 ± 1.1	14.27 ± 0.10	0.81 ± 0.002	0.06 ± 0.003

**Table 6 materials-15-04223-t006:** Values of the potentiodynamic polarization parameters.

Specimen Designation (Type of Surface)	Corrosion Potential E_corr_ (V vs. SCE)	Pitting Potential E_p_ (V vs. SCE)
as received	−0.15 ± 0.01	0.42 ± 0.03
PEP	−0.06 ± 0.02	1.08 ± 0.03
PPEP	0.01 ± 0.02	0.98 ± 0.02

**Table 7 materials-15-04223-t007:** Average corrosion rates calculated from mass losses during the exposure test.

Specimen Designation (Type of Surface)	Average Corrosion Rate (g/(m^2^·day))
as received	0.0011 ± 0.9%
PEP	0.0003 ± 0.8%
PPEP	0.0003 ± 0.8%

## Data Availability

Data sharing is not applicable to this article.

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
