# Peer review of "Plasma Electrolytic Polishing—An Ecological Way for Increased Corrosion Resistance in Austenitic Stainless Steels"

_materials, 2022, doi:10.3390/ma15124223_

Round 1
Reviewer 1 Report
This paper is focused on the corrosion behavior of PEP treated AISI 316L stainless steel. The corrosion resistance of three kinds of samples was evaluated and compared with each other by means of three independent test methods. The authors did a deepgoing research and got some reasonable result. However, some issuses should be resolved before accept.
1. All of the morphology of samples was observed by OM. Maybe some different results would be observed by SEM.
2. The equivalent circuit was too simple for a PEP sample. It should be modified. The relevant equivalent circuit can be cited from Mater. Des. 2022, 215, 110450.
3. The XRD should be added after exposed for a certain time.
4. The conclusion part should be rewrite in a more reasonable way.
Reviewer 2 Report
This manuscript presents the corrosion behavior of PEP treated AISI 316L stainless steel widely used as one of the biomaterials. Corrosion properties of PEP treated surfaces with/without chemical pre-treatment are evaluated and compared with original non-treated (as received) surface by three independent test methods: electrochemical impedance spectroscopy (EIS), potentiodynamic polarization and exposure immersion test. The results obtained indicated high corrosion resistance of PEP treated surfaces also without chemical pretreatment, which increases ecological benefits of PEP technology. My detailed comments are as follows:
1. The decimal places of table 5 and table 6 measurement results are not uniform. And the measurement results of table 7 cannot reach 5 digits after the decimal point. Please check and correct.
2. The black spots in the microstructure diagrams in Figure 2 and figure 8 are caused by corrosion or the second equivalent substance in the structure. Please further clarify the impact on the surface quality.
3. The ruler forms in Figure 1,Figure 2 and Figure 8 are not uniform.
4. In conclusion, plasma electrolytic polishing ensured more than six-fold increase in quality of passive surface film compared to the as received surface. How is sixfold determined? Please elaborate further in the article.
5. What is the prominent point of the blue curve in Figure 6 and why this abnormal point occurs? Please further supplement in the article.
Reviewer 3 Report
Plasma electrolytic polishing - an ecological way for increased 2 corrosion resistance of austenitic stainless steels
The authors describe their work investigating the effect of plasma electrolytic polishing on the corrosion performance of austenitic steel. Unfortunately, the results are hardly compared and a critical discussion is completely missing. In addition, the manuscript needs to be linguistically revised.
1. In the introduction, entire sentences and paragraphs of the abstract are copied 1:1 - here, the effort should be made to elaborate the information in the abstract much more precisely in the introduction. Simple copying of the descriptions is without any additional value.
2. Why is NaCl solution employed in the studies to simulate body-like environmental conditions? Subsequently, a comparison is made with, among others, Hank's solution, whose composition is much closer to the human organism, but which therefore differs significantly from the saline solution. How transferable/comparable are the results then at all? Wouldn't it have been appropriate to have used Hank's solution or other body-like media in the experiments as well?
3. The representation of the electrolyte composition in Table 2 is inappropriate. It is not clear that it is one electrolyte composed of 3 components, which is then used by applying the parameters temperature and time.
4. Please use the wording “surface conditions” instead of “surfaces”, since the conditions are the focal point. (e.g. in the heading of table 3, figure 4)
5. The authors state that three measurements per sample were made to determine the PP curves. Here it remains open whether it is the same sample three times or the same surface condition three times. If assumption one is correct, it must be explained, if it is only one sample, what happens to the surface before the respective subsequent measurement. Particularly since pitting occurs, a better description of the procedure is required here.
6. In Figure 5, the single measuring points of the states "as measured" and " fitted" can hardly be assigned. If applicable, adjust the size of the diagram or at least make it wider.
7. The authors state that for the determination of the corrosion potentials, the table extrapolation cannot be applied (the reason for this remains open) and the respective corrosion potential is derived from the diagram or the respective curve. This must be explained in detail. A simple reading does not seem to be sufficiently objective. Which limits and assumptions are used as a basis? What is the basis for the accuracy of the observed values? The stated accuracies cannot be recognized from the selected representation. Finally, it remains open whether this is an "established procedure".
8. The experimental set-up of the immersion tests does not specify whether the electrolyte solutions are stationary or whether they are in circulation, which would be a much better representation of the human organism. In the context of the overall corrosion tests carried out, it remains open whether buffer solutions were employed to stabilize the pH value, for example, or whether the media in the immersion test were changed or refilled in the meantime. Was saturation or addition of CO2 or O2 implemented? Both gases are found in the human organism and have a significant effect on the degradation mechanism.
9. The majority of the literature sources cited are significantly older than 5 years. There is little reference to more recent studies, important publications are missing.
Finally, some minor issues should be addressed, as well:
a. “austenitic stainless steel biomaterials” is more than one keyword – the authors should either think of a better description or keyword here or at least subdivide these words into 2 or 3 keywords
b. The format of the references given on page 2, line 69 should be corrected.
c. Please use a uniform notation for the specification of the sample sizes (e.g. “15 x 40 mm2“ or “15 mm x 40 mm” or “(15 x 40) mm2”)
d. The scale bars in Figure 2 and Figure 8 are only hardly readable.
e. Please, pay attention to subscripts of the indices also in the figures (Figure 3)

Round 2
Reviewer 3 Report
most comments were addressed quite good, but some minor issues regarding the formatting of tables could be focussed on in the proofreading process.